# WORD SENSE INDUCTION WITH KNOWLEDGE DISTILLATION FROM BERT

## ABSTRACT

Pre-trained contextual language models are ubiquitously employed for language understanding tasks, but are unsuitable for resource-constrained systems. Non-contextual word embeddings are an efficient alternative in these settings. Such methods typically use one vector to encode multiple different meanings of a word, and incur errors due to polysemy. This paper proposes a two-stage method to distill multiple word senses from a pre-trained language model (BERT) by using attention over the senses of a word in a context and transferring this sense information to fit multi-sense embeddings in a skip-gram-like framework. We demonstrate an effective approach to training the sense disambiguation mechanism in our model with a distribution over word senses extracted from the output layer embeddings of BERT. Experiments on the contextual word similarity and sense induction tasks show that this method is superior to or competitive with state-of-the-art multi-sense embeddings on multiple benchmark data sets, and experiments with an embedding-based topic model (ETM) demonstrates the benefits of using this multi-sense embedding in a downstream application.

## 1 INTRODUCTION

While modern deep contextual word embeddings have dramatically improved the state-of-the-art in natural language understanding (NLU) tasks, shallow noncontextual representation of words are more practical solution in settings constrained by compute power or latency. In single-sense embeddings such as `word2vec` or GloVe, the different meanings of a word are represented by the same vector, which leads to the meaning conflation problem in the presence of polysemy. Arora et al. (2018) showed that in the presence of polysemy, word embeddings are linear combination of the different sense embeddings. This leads to another limitation in the word embeddings due to the triangle inequality of vectors (Neelakantan et al., 2014). For *apple*, the distance between the words similar to its two meanings, i.e. *peach* and *software*, will be less than the sum of their distances to *apple*. To account for these and other issues that arise due to polysemy, several multi-sense embeddings model have been proposed to learn multiple vector representations for each word.

There are two major components for a multi-sense embedding model to be used in downstream tasks: (1) *sense selection* to find the correct sense of a word in context (2) *sense representation* for the different meanings of a word. Two types of tasks are used to evaluate these two distinct abilities of a word sense embedding model: word sense induction and contextual word similarity respectively. In word sense induction, a model has to select the correct meaning of a word given its context. For contextual word similarity, the model must be able to embed word senses with similar meaning close to each other in the embedding space. Prior works on multi-sense embeddings have not been shown to be effective on these two tasks simultaneously (e.g. AdaGram (Bartunov et al., 2016) works well on word sense induction but not on contextual word similarity). We propose a method to distill knowledge from a contextual embedding model, BERT (Devlin et al., 2018), for learning a word sense model that can perform both tasks effectively. Furthermore, we show the advantage of sense embeddings over word embeddings for training topic models.

Pre-trained language models like BERT, GPT (Radford et al., 2019), RoBERTa (Liu et al., 2019) etc. have been immensely successful in multiple natural language understanding tasks, but the computational requirements and large latencies of these models inhibits their use in low-resource systems. Static word embeddings or sense embeddings have significantly lower computational

requirements and lower latencies compared to these large transformer models. In this work, we use knowledge distillation (Hinton et al., 2015) to transfer the contextual information in BERT for improving the learning of multi-sense word embedding models. Although there are several prior works on distilling knowledge to smaller transformer models, to the best of our knowledge there has been no work on using knowledge distillation to improve the training of word or sense embeddings using pre-trained language models.

The contributions of this paper are:

   (i) A two-stage method for knowledge distillation from BERT to static word sense embeddings
  (ii) A sense embedding model with state-of-the-art simultaneous performance on sense selection and sense induction.
 (iii) A demonstration of the increased efficacy of using these multi-sense embedding in lieu of single sense in embeddings in a downstream task, namely embedded topic modeling (Dieng et al., 2020).

## 2 RELATED WORK

Here we describe a number of previous models trained to represent different meanings of words using the *word2vec* framework. Prior works on model distillation using BERT are also discussed.

Reisinger & Mooney (2010) was the first to learn vector representations for multiple senses of a word. They extracted feature vectors of a word in different contexts from the context words and clustered them to generate sense clusters for a word. The cluster centroids are then used as sense embeddings or prototypes. Huang et al. (2012) similarly clustered the context representations of a word using spherical k-means, and learned sense embeddings by training a neural language model.

Neelakantan et al. (2014) extended the skip-gram model to learn the cluster centers the sense embeddings simultaneously. At each iteration, the context embedding is assigned to the nearest cluster center and the sense embedding for that cluster is used to predict the context word. A non-parametric method is employed to learn an appropriate number of clusters during training.

Tian et al. (2014) defined the probability of a context word given a center word as a mixture model with word senses representing the mixtures. They developed an EM algorithm for learning the embeddings by extending the skip-gram likelihood function. Li & Jurafsky (2015) used the Chinese Restaurent Process to learn the number of senses while training the sense embeddings. They evaluated their embeddings on a number of downstream tasks but did not find a consistent advantage in using multi-sense embeddings over single-sense embeddings.

Athiwaratkun & Wilson (2017) represented words as multi-modal Gaussian distributions where the different modes correspond to different senses of a word. To denote the similarity between two word distributions, they used the expected likelihood kernel replacing cosine similarity for vectors.

Bartunov et al. (2016) proposed a non-parameteric Bayesian extension to the skip-gram model to learn the number of senses of words automatically. They used the Dirichlet Process for infinite mixture modeling by assuming prior probabilities over the meanings of a word. Due to the intractibility of the likelihood function, they derived a stochastic variational inference algorithm for the model.

Grzegorczyk & Kurdziel (2018) learned sense disambiguation embeddings to model the probability of different word senses in a context. They used the gumbel-softmax method to approximate the categorical distribution of sense probabilities. To learn the number of senses of a word, they added an entropy loss and combined it with a pruning strategy.

Lee & Chen (2017) proposed the first sense embedding model using a reinforcement learning framework. They learned only sense embeddings and an efficient sense selection module for assigning the correct sense to a word in context.

Hinton et al. (2015) demonstrated the effectiveness of employing a large pre-trained teacher network to guide the training of a smaller student network and named this method **model distillation**. Recently, this technique has been used to train many efficient transformer models from BERT. Sun et al. (2019); Sanh et al. (2019); Jiao et al. (2019); Sun et al. (2020) have trained smaller BERT models by distilling knowledge from pre-trained BERT models down to fewer layers. However, there seems to be no prior work on improving word or sense embeddings by using pre-trained language models like BERT.

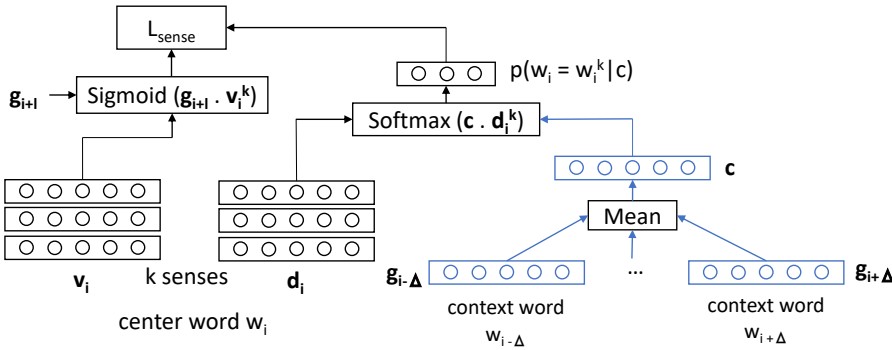

Figure 1: Sense embedding model

# 3 MODEL

We first describe our word sense embedding model. Then the details of extracting knowledge from the contextual embeddings in BERT are discussed.

## 3.1 SENSE EMBEDDING MODEL

We have a dataset $D = \{w_1, w_2, \ldots, w_N\}$, a sequence of $N$ words from a vocabulary $V$. For a word $w_i$ in our vocabulary, we assume $K$ senses and for the $k$-th sense, we learn a sense embedding, $\mathbf{v}_i^k$ and a disambiguation embedding, $\mathbf{d}_i^k$. We also have a global embedding $\mathbf{g}_i$ for each word $w_i$.

Following the `word2vec` framework to train the sense embeddings, we slide a window of size $2\Delta + 1$ (center word and $\Delta$ words to its left and right) over the dataset. In a context window $C = \{w_{i-\Delta}, \ldots, w_i, \ldots, w_{i+\Delta}\}$, we define the conditional probability of a context word given the center word, $w_i$ as

$$p(w_{i+l}|w_i) = \sum_{k=1}^{K} p(w_{i+l}|w_i = w_i^k)p(w_i = w_i^k|w_i, C) \tag{1}$$

where, $l \in [-\Delta, \Delta], l \neq 0$.

Here the probability of a context word, $w_{i+l}$ given a sense embedding, $w_i^k$ is

$$p(w_{i+l}|w_i = w_i^k) = \sigma(\langle \mathbf{g}_{i+l}, \mathbf{v}_i^k \rangle) \tag{2}$$

where $\sigma(x) = (1 + \exp(-x))^{-1}$ is the logistic sigmoid function.

The sense disambiguation vector selects the more relevant sense based on the context. The probability of sense, $k$ of word, $w_i$ in context $C$ is defined similar to the dot-product attention mechanism.

$$p(w_i = w_i^k|w_i, C) = \frac{\exp(\langle \mathbf{d}_i^k, \mathbf{c} \rangle)}{\sum_{k=1}^{K} \exp(\langle \mathbf{d}_i^k, \mathbf{c} \rangle)} \tag{3}$$

where, $\mathbf{c}$ is the context embedding for context $C$ defined by the average of the global embeddings of the context words

$$\mathbf{c} = \frac{1}{2\Delta} \sum_{\substack{l=-\Delta \\ l \neq 0}}^{\Delta} \mathbf{g}_{i+l} \tag{4}$$

**Iterative sense disambiguation** Instead of using the global embeddings of the context words, we can use their sense embeddings and sense probabilities to obtain context embedding with more contextual information. The steps are following:

**Step 1** Using the context embedding from Eq. 4, compute the sense probability $p(w_{i+l} = w_{i+l}^k|w_{i+l}, C)$ for each word in the context ($l \in [-\Delta, \Delta], l \neq 0$)

**Step 2** Replace the global embeddings in Eq. 4 with the weighted average of the sense embeddings for each context word

$$\mathbf{c} = \frac{1}{2\Delta} \sum_{\substack{l=-\Delta \\ l \neq 0}}^{\Delta} \sum_{k=1}^{K} \mathbf{v}_{i+l}^{k} \, p(w_{i+l} = w_{i+l}^{k} | w_{i+l}, C) \tag{5}$$

Similar to the skip-gram model (Mikolov et al., 2013), we employ the negative sampling method to reduce the computational cost of a softmax over a huge vocabulary. The loss function for the multi-sense embedding model is as follows

$$L_{\text{sense}} = - \sum_{w_i \in D} \sum_{\substack{l=-\Delta \\ l \neq 0}}^{\Delta} \left\{ \log[p(w_{i+l} | w_i)] + \sum_{j=1}^{n} \log[1 - p(w_{i+l,j} | w_i)] \right\} \tag{6}$$

where, n is the number of negative samples and $w_{i+l,j}$ is the $j$-th negative sample for the context word $w_{i+l}$.

## 3.2 TRANSFER KNOWLEDGE FROM BERT

We use the contextual embeddings from BERT to improve our sense embedding model in 2 steps. First, we learn the sense embeddings as a linear decomposition of the BERT contextual embedding in a context window. Second, the sense embeddings from the first step are used for knowledge distillation to our sense embedding model.

### 3.2.1 BERT SENSE MODEL

Let, the sense embeddings of word $w_i$ trained from BERT is $\{\mathbf{u}_i^1, \ldots, \mathbf{u}_i^K\}$ and the contextual embedding for word, $w_i$ is $\mathbf{b}_i$ - embedding from the output layer of BERT. Since BERT uses WordPiece tokenization, some words are divided into smaller tokens. So each word has multiple 'wordpiece' tokens and output embeddings. To get the contextual embedding $\mathbf{b}_i$ from these token embeddings, we take the mean of all token output embeddings for a word. Similar to Eq. 3 the probability of sense, $k$ of word, $w_i$ in context $C$ is

$$p(w_i = w_i^k | w_i, C) = \frac{\exp(\langle \mathbf{d}_i^k, \mathbf{c}_{\text{BERT}} \rangle)}{\sum_{k=1}^{K} \exp(\langle \mathbf{d}_i^k, \mathbf{c}_{\text{BERT}} \rangle)} \tag{7}$$

where, context embedding

$$\mathbf{c}_{\text{BERT}} = \frac{1}{2\Delta} \sum_{l=-\Delta}^{\Delta} \mathbf{b}_{i+l}$$

We want the weighted sum of the sense embeddings to be similar to the contextual embedding from BERT. So, the loss function is

$$L_{\text{BERT}} = \sum_{w_i \in D} \left\| \sum_{k=1}^{K} p(w_i = w_i^k | w_i, C) \mathbf{u}_i^k - \mathbf{b}_i \right\|_2^2 \tag{8}$$

### 3.2.2 KNOWLEDGE DISTILLATION FROM BERT

Here we combine our multi-sense embedding model and the sense embeddings from BERT to transfer the contextual information. To be precise, we want to improve the sense disambiguation module (3) of the multi-sense embedding model using BERT. The key idea of knowledge distillation is that the output from a complex network contains more fine grained information than a class label. So we want the student network to produce similar probabilities as the teacher network. Hinton et al. (2015) suggested using a "softer" version of the output from the teacher network using a temperature $T$ over the output logits $\mathbf{z}_i$:

$$p^T(x) = \frac{\exp(\mathbf{z}_i / T)}{\sum_i \exp(\mathbf{z}_i / T)}$$

where, $x$ is the input.

The loss for the knowledge transfer is the combination of the original loss (Eq. 6) and the cross-entropy loss between the sense disambiguation probability from the sense embedding (Eq. 3) model and the BERT model (Eq. 7).

$$L = L_{\text{sense}} + \alpha L_{\text{transfer}} \tag{9}$$

where

$$L_{\text{transfer}} = -T^2 \sum_{k=1}^{K} p_{\text{sense}}^T(w_i = w_i^k | w_i, C) \log(p_{\text{BERT}}^T(w_i = w_i^k | w_i, C)) \tag{10}$$

is the cross-entropy loss between the distilled sense distribution and the BERT sense distribution. We multiply this loss by $T^2$ because the magnitudes of the gradients in the distillation loss scale by $1/T^2$.

## 4 EXPERIMENTS

We conduct both qualitative and quantitative experiments on our trained sense embedding model.

### 4.1 DATASET AND TRANING

We trained our models on the April 2010 snapshot of Wikipedia (Shaoul & Westbury, 2010) for fair comparison with the prior multi-sense embedding models. This corpus has over 2 million documents and about 990 million words. We also used the same vocabulary set as Neelakantan et al. (2014).

We selected the hyperparameters of our models by training them on a small part of the corpus and using the loss metric on a validation set. For training, we set the context window size to $\Delta = 5$. We use 10 negative samples for each positive (word, context) pair. We do not apply subsampling like `word2vec` since the BERT model was pre-trained on sentences without subsampling. We learn 3 sense embeddings for each word with a dimension of 300 for comparison with prior work. We use the Adam (Kingma & Ba, 2014) optimizer with a learning rate of 0.001.

The base model of BERT is used as the pre-trained language model. The PyTorch *transformers* (Wolf et al., 2019) library is used for inference through the BERT-base model. With a batch size of 2048, it takes 10 hours to train the sense embedding model on the wikipedia corpus for one epoch in a Tesla V100 GPU. The sense embedding models with and without knowledge distillation are trained for 2 epochs over the full Wikipedia dataset. The weight for the knowledge distillation loss (Eq. 9) $\alpha = 1$ is selected from {0.1, 0.5, 1.0}. Temperature parameter $T$ is set to 4 by tuning over the range [2,6].

### 4.2 NEAREST NEIGHBOURS

The nearest neighbours of words in context are shown in Table 1. Given the context words, the context embedding is computed using Eq. 5 and sense probability distribution is computed with Eq. 3. The most probable sense is selected and then the cosine distance is computed with the global embeddings of all words in the vocabulary. Table 1 also shows the values of the sense probability for the word in context. The sense probabilities are close to 1 in most of the cases. This indicates that the model learned a narrow distribution over the senses and it selects a single sense most of the time.

### 4.3 WORD SENSE INDUCTION

This task evaluates the model's ability to select the appropriate sense given a context. To assign a sense to a word in context, we take the words in its context window and compute the context embedding using Eq. 5. Then the sense probability from Eq. 3 gives the most probable sense for that context.

We follow Bartunov et al. (2016) for the evaluation dataset and metrics. Three WSI datasets are used in our experiments. The SemEval-2007 dataset is from SemEval-2007 Task 2 competition (Agirre & Soroa, 2007). It contains 27232 contexts for 100 words collected from Wall Street Journal (WSJ) corpus. The SemEval-2010 dataset is from the SemEval-2010 Task 14 competition (Manandhar &

| Context | Sense: Prob. | Nearest Neighbours |
|---|---|---|
| macintosh is a family of computers from *apple* | 1 : 0.99 | apple, amstrad, macintosh, commodore, dos |
| *apple* software includes mac os x | 2 : 0.963 | microsoft, ipod, announced, software, mac |
| *apple* is good for health | 3 : 0.999 | apple, candy, strawberry, cherry, blueberry |
| the power *plant* is closing | 3 : 1.0 | mill, plant, cogeneration, factory, mw |
| the science of *plant* life | 2 : 1.0 | species, plant, plants, lichens, insects |
| the seed of a *plant* | 1 : 0.99 | thistle, plant, gorse, salvia, leaves |
| the *cell* membranes of an organism | 3 : 1.0 | depolarization, contractile, hematopoietic, follicular, apoptosis |
| almost everyone has a *cell* phone now | 2 : 0.99 | phone, cell, phones, mobile, each |
| prison *cell* is not a pleasant place | 1 : 0.99 | prison, room, bomb, arrives, jail |

Table 1: Top 5 nearest neighbours of words in context based on cosine similarity. We also report the index of the selected sense and its probability in that context.

Klapaftis, 2009) which contains 8915 contexts for 100 words. Finally, the SemEval-2013 dataset comes from the SemEval-2013 Task 13 (Jurgens & Klapaftis, 2013) with 4664 contexts for 50 words.

For evaluation, we cluster the different contexts of a word according to the model's predicted sense. To evaluate the clustering of a set of points, we take pairs of points in the dataset and check if they are in the same cluster according to the ground truth. For the SemEval competitions, two traditional metrics were used: F-score and V-measure. But there are intrinsic biases in these methods. F-score will be higher for a cluster with fewer centers while V-measure will be higher for clusters with a large number of centers. So Bartunov et al. (2016) used ARI (Adjusted Rand Index) as the clustering evaluation method since it does not suffer from these problems.

We report the ARI scores for MSSG (Neelakantan et al., 2014), MPSG (Tian et al., 2014) and AdaGram (Bartunov et al., 2016) from Bartunov et al. (2016). Scores for disambiguated skip-gram are collected from Grzegorczyk & Kurdziel (2018). We used the trained embeddings published by MUSE Lee & Chen (2017) with Boltzmann method for this task. Single-sense models like skip-gram cannot be used in this task since the task is to assign different senses to different contexts of a word. Contextual models like BERT give an embedding for a word given its context, but this embedding is not associated with a specific discrete meaning of the word. Therefore BERT models are not suitable for comparison on this task.

MUSE shows strong performance in word similarity tasks (section 4.4) but fails to induce meaningful senses given a context. The Disambiguated Skip-gram model was the best model on SemEval-2007 and SemEval-2010 datasets. Our sense embedding model performs better than MSSG and MPSG models but there is a large gap with the state-of-the-art. We can see a significant improvement in ARI score ($1.25x - 2.25x$) on the 3 datasets with knowledge distillation from BERT.

| Model | SE-2007 | SE-2010 | SE-2013 |
|---|---|---|---|
| MSSG | 0.048 | 0.085 | 0.033 |
| MPSG | 0.044 | 0.077 | 0.033 |
| AdaGram | 0.069 | 0.097 | 0.061 |
| MUSE | 0.0009 | 0.01 | 0.006 |
| Disamb. SG | 0.077 | 0.117 | 0.045 |
| SenseEmbed | 0.053 | 0.076 | 0.054 |
| BERTSense | 0.179 | 0.179 | 0.155 |
| BERTKDEmbed | **0.145** | **0.144** | **0.133** |

Table 2: ARI (Adjusted Rand Index) scores for word sense induction tasks. All models have dimension of 300. The Adagram and Disambiguated skip-gram models learn variable number of senses while all other models learn fixed number of senses per word. All baseline results except MUSE were collected from Grzegorczyk & Kurdziel (2018). SenseEmbed - Sense embedding model without distillation, BERTSense - Sense embedding model (Sec. 3.2.1), BERTKDEmbed - Sense embedding with distillation

## 4.4 CONTEXTUAL WORD SIMILARITY

Here, the task is to predict the similarity of the meaning of 2 words given the context they were used. We use the Stanford Contextual Word Similarity (SCWS) dataset (Huang et al., 2012). This dataset

contains 2003 word pairs with their contexts, selected from Wikipedia. Spearman's rank correlation $\rho$ is used to measure the degree of agreement between the human similarity scores and the model's similarity scores.

We use the following methods for predicting similarity of two words in context for a sense embedding model following Reisinger & Mooney (2010): AvgSimC and MaxSimC. The AvgSimC measure weighs the cosine similarity of all combinations of sense embeddings of 2 words with the probability of those 2 senses given their respective contexts.

$$\text{AvgSimC}(w_1, w_2) = \frac{1}{K^2} \sum_{i=1}^{K} \sum_{j=1}^{K} p(w_1 = w_1^i | w_1, C) p(w_2 = w_2^j | w_2, C) \cos(v_1^i, v_2^j) \quad (11)$$

Here, the cosine similarity between sense $i$ word 1 and sense $j$ of word 2 is multiplied by the probability of these 2 senses given their contexts.

The MaxSimC method selects the best sense for each context based on their probabilities and measures their cosine similarity.

$$\text{MaxSimC}(w_1, w_2) = \cos(v_1^i, v_2^j) \quad (12)$$

where

$$i = \arg \max_{k=1}^{K} p(w_1 = w_1^k | w_1, C) \quad \text{and} \quad j = \arg \max_{k=1}^{K} p(w_2 = w_2^k | w_2, C) \quad (13)$$

For comparison with baselines on this task, we use the single-sense skip-gram model (Mikolov et al., 2013) with 300 and 900 dimensions since our word sense model has 3 embeddings per word with 300 dimensions each. We use the contextual models (BERT, DistilBERT) by selecting the token embedding from the output layer for the word in a context window of 10 words to the left and right.

We report the score for the best performing model from each method in Table 3. For MUSE, the best model (Boltzmann method) is used. [1]. Our sense embedding model (SenseEmbed) is competitive in AvgSimC measure whereas the knowledge distillation method significantly improves the MaxSimC score. We can infer from this result that the improvement in sense induction mechanism helped learn better embeddings for each word sense. The best performing multi-sense embedding model's performance is very close to the 900 dimensional skip-gram embeddings (69.3 vs 68.4). The best performing multi-sense model considering the average over the 2 measures is MUSE with an average of 68.2. But it does not do well on the word sense induction task. Both contextual embedding models (BERT and DistilBERT (Sanh et al., 2019)) obtained lower scores than our sense embedding model.

| Model | AvgSimC | MaxSimC |
|---|---|---|
| Skip-Gram (300D) | 65.2 | 65.2 |
| Skip-Gram (900D) | 68.4 | **68.4** |
| MSSG | **69.3** | 57.26 |
| MPSG | 65.4 | 63.6 |
| AdaGram | 61.2 | 53.8 |
| MUSE | 68.8 | 67.6 |
| BERT-base | 64.26 | 64.26 |
| DistilBERT-base | 66.51 | 66.51 |
| SenseEmbed | 67.6 | 54.7 |
| BERTSense | 62.6 | 64 |
| BERTKDEmbed | 68.6 | 67.2 |

Table 3: Spearman rank correlation score ($\rho \times 100$) on the SCWS dataset. Skip-gram is the only single-sense embedding model. All other embedding models are 300 dimensional. Skip-gram results are reported in Bartunov et al. (2016), MUSE score is obtained using their published model weights and for other models the results are from the respective papers.

### 4.5 Effectiveness of Knowledge Distillation

We can use the intermediate sense embeddings $\{\mathbf{u}_i^1, \ldots, \mathbf{u}_i^K\}$ from the BERT Sense model (Sec. 3.2.1) on word sense induction and contextual word similarity tasks. Since the BERT sense embeddings are trained using the output embeddings from BERT, we can use them as context embedding for sense induction.

---

[1] https://github.com/MiuLab/MUSE, we used trained embeddings published here

In the knowledge distillation method, we train our sense embedding model to mimick the sense distribution from the BERT Sense model. Since the WSI task evaluates the predicted sense of the model, this is the ideal task to quantify the quality of knowledge transfer from BERT to the sense embedding model. The BERT Sense model is compared with the sense embedding model trained with knowledge distillation in Table 4. As the BERT Sense model is trained with MSE (Mean Squared Error) loss, it's sense embeddings have the same dimensionality as BERT (768). From Table 4, we see that the 300 dimensional sense embedding model can obtain $75\%$ to $88\%$ ARI scores without using the 12 layer transformer model with 768 dimensional embeddings.

| Model | SE-2007 | SE-2010 | SE-2013 |
|---|---|---|---|
| BERTSense | 0.195 | 0.163 | 0.155 |
| BERTKDEmbed | **0.145** | **0.144** | **0.133** |

Table 4: Word sense induction using the BERT Sense model (768D) and knowledge distilled sense embedding model (300D). ARI (Adjusted Rand Index) scores are reported.

| Model | AvgSimC | MaxSimC |
|---|---|---|
| BERTSense | 62.6 | 64 |
| BERTKDEmbed | 68.6 | 67.2 |

Table 5: Comparison of BERT sense model with sense embedding trained with knowledge distillation. Spearman rank correlation ($\rho \times 100$) scores are reported on the SCWS dataset.

We also compare the BERT sense model with the sense embedding model on the contextual word similarity task in Table 5. The lower correlation score from BERT Sense model indicates that these embeddings are not better at sense representation although they learn to disambiguate senses. The loss function in Eq. 8 helps the BERT Sense model to extract contextual sense information in BERT output embeddings, but it is not optimal for training sense embeddings. The sense embedding model combined with the knowledge gained from the BERT sense model learns to do both i) sense selection and ii) sense representation effectively.

## 4.6 EMBEDDED TOPIC MODEL

As sense embeddings help discern the meaning of a word in context, prior work has shown the benefits of using them to perform topic modeling in the embedding space (Dieng et al., 2020). We explore the further benefits of using embeddings which model the fact that one token can have multiple meanings. Given a word embedding matrix $\rho$ where the column $\rho_v$ is the embedding of word $v$, Embedded Topic Model (ETM) follows the generative process:

1. Draw topic proportions $\theta_d \sim \mathcal{LN}(0, I)$
2. For each word $n$ in the document:
   (a) Draw topic assignment $z_{dn} \sim \text{Cat}(\theta_d)$
   (b) Draw the word $w_{dn} \sim \text{softmax}(\rho^T \alpha_{z_{dn}})$

Here, $\alpha_k$ is the topic embedding for topic $k$. These topic embeddings are used to obtain per-topic distribution over the vocabulary of words.

We experimentally compare the performance of ETM models trained[2] using skip-gram word embeddings, our sense embeddings and MSSG (Neelakantan et al., 2014) embeddings. Three senses were trained for each word in both of the multi-sense models. A 15K document training set, 4.9K test set, and 100 document validation set are sampled from the Wikipedia data set and used for the experiments. During training using our sense embeddings and the MSSG multi-sense embeddings, the sentences are represented using a "bag-of-sense" representation where each word is replaced with its most probable sense , selected using a context window of 10 words; with the skip-gram embeddings, bag-of-words representations are used.

For evaluation, we measured the document completion perplexity, topic coherence, and topic diversity for different vocabulary sizes. Document perplexity measures how well the topic model can be used

---

[2]Using code published by the authors of the original ETM model; see `https://github.com/adjidieng/ETM`.

to predict the second half of a document, given its first half. Topic coherence measures the probability of the top words in each topic to occur in the same documents, while topic diversity is a measure of the uniqueness of the most common words in each topic. Higher topic coherence and topic diversity indicate that a model performs better at extracting distinct topics; the product of the two measures is used as a measure of topic quality:

$$\text{topic quality} = \text{topic coherence} \times \text{topic diversity}$$

We follow (Dieng et al., 2020) in reporting the ratio of perplexity and topic quality as a measure of the performance of the topic model: a lower ratio is better.

We set the minimum document frequency to [5, 10, 30, 100] to change the vocabulary size. ETM is initialized with pre-trained sense embeddings and then fine tuned along with the topic embeddings. We followed the hyperparameters reported in Dieng et al. (2020) to train all models. Since we use a bag-of-sense model, ETM provides a probability distribution over the senses. We sum the probability for the different senses of a word to obtain the probability for a word. In Figure 2, we show the ratio of the document completion perplexity (normalized by vocabulary size) and the topic quality.

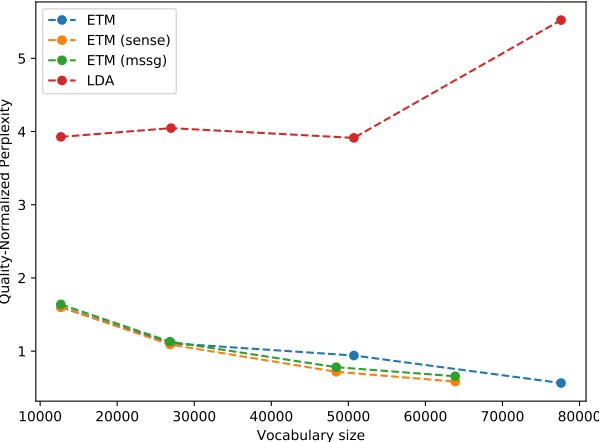

Figure 2: Comparison of word embeddings and sense embeddings for topic modeling with different vocabulary sizes. Here, ETM (sense) - ETM with BERTKDEmbed, ETM (mssg) - ETM with MSSG.

We see that both MSSG and BERTKDEmbed ETM models obtain lower perplexity-quality ratio for higher vocabulary sizes compared to the single-sense (skip-gram) ETM model. This confirms that multi-sense embedding models produce Embedded Topic Models with higher performance. BERTKDEmbed ETM also outperforms the other multi-sense ETM model.

## 5    CONCLUSION

We present a multi-sense word embedding model for representing the different meanings of a word using knowledge distillation from a pre-trained language model (BERT). Unlike prior work on model distillation from contextual language models, we focus on extracting non-contextual word senses; this is accomplished in an unsupervised manner using an intermediate attention-based sense disambiguation model. We demonstrate start-of-the-art results on three word sense induction data sets using fast and efficient sense embedding models. This is the first multi-sense word embedding model with strong performance on both the word sense induction and contextual word similarity tasks. As a downstream application, we show that the sense embedding model increases the performance of the recent embedding-based topic model (Dieng et al., 2020). In future, we will explore the benefits of using pre-trained multi-sense embeddings as features for other downstream tasks, and tackle the task of tuning the number of word meanings for each individual word.

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
