# OpenReview forum: "Word Sense Induction with Knowledge Distillation from BERT"
_ICLR.cc/2022/Conference — ICLR 2022 Submitted_

### Official Review · Reviewer_oDq7 · 2021-11-01

**Correctness:** 2
**Technical Novelty And Significance:** 3
**Empirical Novelty And Significance:** 3
**Recommendation:** 3
**Confidence:** 4

**Main Review:**

This paper presents a multi-sense word embedding model for polysemy words by distilling knowledge from the contextual word embedding model (BERT). The authors first incorporate an attention mechanism over the senses of a word into a skip-gram-like framework. Then, BERT-enhanced sense embeddings are used to improve the skip-gram framework by knowledge distillation. In experiments, the authors confirm the effectiveness of the sense embedding model on three tasks: word sense induction, contextual word similarity, and embedded topic model.

One limitation of the proposed method is the assumption that one word has a fixed number (three in the paper) of word senses, restricting its effectiveness and generalization in downstream tasks. I wonder why the authors chose three and how this would influence the WSI task if a word has more or less sense than 3 in reality?

There have been studies to perform word sense disambiguation with pre-trained models and sense inventory [1,2,3]. Without comparison and analysis of these critical works, it is not easy to verify the technical contribution of this paper. This is important since the proposed technical design (e.g., the attentional skip-gram model) is simple, though I do not see simplicity as a shortcoming.

The performance improvement on the contextual word similarity task looks marginal. Regarding the word induction task, BERT-based models perform much better than non-BERT-based models. The authors are encouraged to do some quantitative analysis on the time-consuming on BERTSense and BERTKDEmbed. Meanwhile, comparing further with some other existing distilled (or compressed) BERT models here will highlight BERTKDEmbed's effectiveness and efficiency.

I find the overall training process is somewhat hard to follow. The clarity of the whole method can be improved, and some detailed design seems not well-motivated. For example, why are two individual embeddings d_i and v_i used in sense representation and disambiguation, respectively? In some similar works [3], coordinating sense and context embeddings is enough to perform disambiguation. In Sec. 3.2.1, which parameters are trainable? Is D^i_k here shared with the D^i_k  in Sec. 3.1?

[1] GlossBERT: BERT for Word Sense Disambiguation with Gloss Knowledge. EMNLP 2019

[2] Analysis and Evaluation of Language Models for Word Sense Disambiguation. Computational Linguistics 2021

[3] Leveraging Human Prior Knowledge to Learn Sense Representations. ECAI 2020


**Summary Of The Paper:**

This paper proposes a two-stage method to distill word sense knowledge from BERT into a skip-gram-like framework. The authors show that the proposed method performs well on sense selection and induction (or representation). They also verify the efﬁcacy on the downstream task of embedded topic modeling.

**Summary Of The Review:**

The effectiveness and generalization of the proposed method are somewhat limited due to the fixed sense number. The novelty seems not strong, and missing high-related works make this paper difficult to position in the research background. The clarity also has room for improvement.

---

> ### Author Response · Authors · 2021-11-19
> **Choice of number of senses and explanation of the notations**
>
> The number of senses for a word is a hyperparameter for our model. With three senses, we see the benefits of allowing for multiple senses, and we outperform or perform on par with prior state-of-the-art methods, even ones that adaptively choose word senses.  Using external knowledge bases to get the number of senses isn’t informative, as the number of meanings that a word has depends on the context and level of distinction relevant to the tasks at hand. The question of choosing an appropriate number of senses to maximally improve performance for individual words on a given task is interesting and a subject of potential future work.
>
> Here is the comparison of the inference time of the proposed models. We can see there is a huge improvement in inference speed going from BERT to word sense embedding.
>
> Model ------------------- Inference time / example
>
> MUSE ------------------ 0.06 ms
>
> BERTKDEmbed ----- 0.06 ms
>
> BERTSense ---------- 32.35 ms
>
>
>
> We separate the sense embeddings v_i and the disambiguation embeddings d_i to ensure the different senses do not collapse into each other. The sense embedding model in [3] computes a contextual representation of each sense for disambiguation using word co-occurrence count. Using the same vector for both representation and disambiguation would make it difficult to learn the sense probabilities.
> In section 3.2.1, we train a separate model with sense embeddings and disambiguation embeddings. So, the disambiguation embedding, d_i, in this section is different from the disambiguation embedding in section 3.1. We will clarify this in our revision. The contextual embedding obtained from BERT is kept fixed during training.

---

### Official Review · Reviewer_pvxe · 2021-11-01

**Correctness:** 3
**Technical Novelty And Significance:** 3
**Empirical Novelty And Significance:** 3
**Recommendation:** 6
**Confidence:** 3

**Main Review:**

Strengths
This work is able to leverage the contextual knowledge of BERT to improve the performance of non-contextual word sense induction.
The performance of the proposed methods on downstream sense induction and word similarity is comparable to those of existing SoTA methods; The distilled sense embedding model yields lower topic perplexity in embedded topic modeling.
The combined worse sense and BERT distillation loss force the resulting distilled model to be performant in both word sense disambiguation and induction.

Concerns
The authors propose static multisense embedding distillation as a solution to the latency concerns that emerge from running large-scale BERT models. There are no experiments showing the computational overhead of their proposed method.
A set of nearest neighbor results shown in sec4.2 are used to suggest that the sense disambiguation network largely selects a single word sense. A stronger justification of this claim would be to show the distribution of sense probabilities computed over all examples rather than in selected examples.

Suggestions & Nit:
Would like to see a Related Work discussion of contextual word sense induction models (especially those using BERT-style transformers).
https://arxiv.org/pdf/1909.08358.pdf
https://arxiv.org/pdf/1908.07245.pdf
Table 2/4: Bolding of the proposed method is misleading as it is not the most performant model. In contrast, highest accuracy is highlighted in Table 3.


**Summary Of The Paper:**

Single-sense word representations which collapse multiple word meanings into a single vector representation are often used in settings where large model inference is impractical. The authors propose a method for extracting noncontextual multiple sense embeddings from transformer-based encoder models via knowledge distillation.

The author’s model predicts likelihoods for the different senses of a target word as the combination of a learned disambiguation vector for each sense and a context embedding constructed from a sliding window over the surrounding words (a-la skipgram). The output multi-sense embedding is the probability weighted sum of the individual sense embeddings. A distillation loss learning against the BERT sense likelihoods is used to improve disambiguation.


**Summary Of The Review:**

Overall Recommendation: Lean Accept
The paper proposes a new noncontextual model for handling multiple word senses by constructing a sense embedding model with improved disambiguation learned via BERT knowledge distillation. As compared to existing work, this work is able to both perform word sense induction while also having strong performance on word similarity. This suggests that the model is able to both disambiguate and also learn strong underlying representations.

I would like to see my above concerns addressed by the authors. For example, it would be beneficial to report the latency of model inference and parameter counts (as relevant) for the proposed sense model, BERT, and other baselines averaged over the test/val set in the word sense induction and word similarity tasks. Likewise re:Concern #2, I would like to see a histogram over the top word sense probabilities.

---

> ### Author Response · Authors · 2021-11-19
> **Inference time and sense distribution**
>
> The comparison of inference time between the sense embedding models and the BERT model.
>
> Model ------------------- Inference time / example
>
> MUSE ------------------ 0.06 ms
>
> BERTKDEmbed ----- 0.06 ms
>
> BERTSense ---------- 32.35 ms
>
> We can see that there is a huge difference in the latency between transformer models and simple word embeddings.
>
> Now, we show the distribution of the sense probabilities for the word "plant" computed over 10,000 examples.
>
> https://ibb.co/album/rkFvGW
>
> The distributions are concentrated around 1 and 0 for all senses indicating that the model assigns a high probability to the most probable sense.

---

### Official Review · Reviewer_2C3z · 2021-11-02

**Correctness:** 3
**Technical Novelty And Significance:** 1
**Empirical Novelty And Significance:** 2
**Recommendation:** 5
**Confidence:** 5

**Main Review:**

Strength:

The writing of the paper is easy to follow. And the paper makes a good attempt to leverage the pre-trained models.

Weakness:

1. One of the main weaknesses is that the results are not significant enough. In Table 3, the performance of the proposed model on both criteria is worse than the existing models. And the paper didn't compare with all existing sense models (see missing citations). In Figure 2, most of the improvements are brought by the ETM model, the benefits of using sense embeddings seem to be very marginal. The paper should also compare with more existing sense embeddings rather than just MSSG. The main improvements that the proposed model has compared to previous sense embeddings are on SemEval tasks, which are not supervised given that contextualized embeddings and pre-trained models perform well on these tasks themselves. The motivation of having sense embeddings is less strong (https://arxiv.org/abs/1909.10430, https://aclanthology.org/2020.coling-main.107.pdf). And the proposed distilled sense embedding performs worse than the BertSense itself. There's no reason for using the distilled embeddings rather than the BERT embeddings themselves.

2. Another main weakness is the novelty of the technique of this paper. Equations 1,2,3,4 have been commonly used in the previous work started by Šuster et al. (2016). The distillation technique is also not novel.

Minor weakness:

1. SemEval is most commonly evaluated by Precision and Recalls, while in this paper, the ARI is used. It's better to add more explanation of what ARI is and why it's used.
2. Missing citations:

Qiu et al. "Context-dependent sense embedding." Association for Computational Linguistics (ACL), 2016
Šuster et al. (2016). Bilingual learning of multi-sense embeddings with discrete autoencoders. In NAACL
Guo et al. "Which Evaluations Uncover Sense Representations that Actually Make Sense?." Proceedings of the 12th Language Resources and Evaluation Conference. 2020.


**Summary Of The Paper:**

This paper proposes a sense embedding model that distills the sense induction probability conditional on context from pre-trained BERT. Experimental results on the word similarity tasks show that it is able to perform well on the word sense induction task and achieve a balance between the contextual and non-contextual word similarity tasks. An experiment on one downstream task, topic modeling, shows that the Embedded Topic Model with the proposed BERT distilled sense embedding is able to decrease the perplexity marginally.

**Summary Of The Review:**

The technique adopted in this paper is simple and not novel. The improvements are not significant.

---

> ### Author Response · Authors · 2021-11-19
> **Semantic similarity performance and novelty in the distillation framework**
>
> 1. Our proposed sense embedding model performs comparably to the other multi-sense embeddings on this task. In particular, the performance of our model in the MaxSimC criteria is close to the best performing model where other models have low scores. Since the goal of sense embeddings is to assign the correct meaning to a word, the MaxSimC measure is more relevant to our task. As the higher dimensional single-sense embedding (900) performs better on this task, we believe the word sense induction task is a better evaluation criteria for multi-sense embedding models.
>
>     As we are trying to match the predicted probabilities of the BERT model with distillation, it is expected that there will be some drop in performance in the distilled model. The motivation for using sense embeddings in place of BERT models is to lower computational overhead during inference.
>
> 2. Our sense embedding model is based on the skip-gram framework. So there is some similarity with previous sense models in the structure. We introduced an iterative sense disambiguation that improves the performance of the model on word sense induction.
> We adapted the original distillation technique to distil word sense information in pre-trained language models by developing a 2 step process. In the first stage, we train an intermediate sense embedding model by using BERT embeddings as context and then we use the sense probabilities learned by this model to train our sense embedding model using global word vectors to represent the context. As far as we know, model distillation has not been used to train single or multi-sense word embeddings before.
>
> Evaluation:
> We follow Bartunov et. al (2016) in using ARI as the cluster evaluation metric rather than the F-score and V-measure used in SemEval, because the former is artificially higher for clusters with few centers while the latter is artificially higher for clusters with a large number of centers.
>
> We will cite the mentioned papers in our revision. We thank you for pointing us to these relevant works.

---

### Official Review · Reviewer_npDw · 2021-11-02

**Correctness:** 4
**Technical Novelty And Significance:** 4
**Empirical Novelty And Significance:** 2
**Recommendation:** 6
**Confidence:** 3

**Details Of Ethics Concerns:**

The submitted paper might violate anonymity because authors are visible through the footnote link to their GitHub account as mentioned in main review.

**Main Review:**

Strength: Their paper is easy to follow. Surveys on the area are quite decent in the related work section. An idea of obtaining multi-sense word embedding is novel. In the word sense induction task, their method exhibits better performance than conventional ones.

Weakness:
Results of their proposed method in contextual word similarity tasks do not outperform existing methods.
Readers cannot know the method’s uniqueness. In particular, semantic similarity task is not supporting this method’s superiority. Discussion of comparison between their proposed method and existing methods is not provided.
Another concern is that authors are visible through the footnote link to their GitHub account.

Minor point
Why is BERTKDEmbed highlighted in Table2?

**Summary Of The Paper:**

The authors proposed a method to obtain suitable multi-sense word embedding based on skip-gram and BERT distillation. In my opinion, from technological point of view, their idea is very novel. They also incorporated their word embedding into embedded topic modeling.

**Summary Of The Review:**

The authors build a new type of obtaining multi-sense word embedding is novel. Although the model is just comparable to existing methods in contextual word similarity tasks, the model and derivatives show potential for both the word sense induction task and the embed topic model. Thus, I found this paper is marginally significant.

---

> ### Comment · Area_Chair_qUDs · 2021-11-16
> **GitHub**
>
> Dear reviewer,
>
> I checked the submission and the two GitHub links provided in footnotes 1 & 2 are for prior work and not for the new contributions in this paper. Were you referring to these GitHub links in your anonymity violation concerns or to something else? If you were referring to these links, I believe there is no anonymity violation.
>
> Thanks,
> Your Area Chair

---

> > ### Comment · Reviewer_npDw · 2021-11-16
> > **I agree with the comment.**
> >
> > Dear Area Chair,
> >
> > Thank you for the comment.
> > I've learned the link doesn't show authors' affiliation.
> > Now I believe there is no violation.
> >
> > Best regards,
> > Anonymous reviewer

---

> ### Author Response · Authors · 2021-11-19
> **Semantic similarity performance**
>
> Our proposed sense embedding model performs comparably to the other multi-sense embeddings on this task. In particular, the performance of our model in the MaxSimC criteria is close to the best performing model where other models have low scores. Since the goal of sense embeddings is to assign the correct meaning to a word, the MaxSimC measure is more relevant to our task. Also the higher dimensional single-sense embedding (900) performs better on this task, so we believe the word sense induction task is a better evaluation criteria for multi-sense embedding models.
>
> The BERTKDEmbed model was highlighted in Table 2 as it is the performing word sense embedding model. The BERTSense model uses BERT to obtain the context embedding.
>
> We would like to reiterate that there was no github link to our account. We thank you for your review.

---

### Official Review · Reviewer_xBJd · 2021-11-03

**Correctness:** 3
**Technical Novelty And Significance:** 3
**Empirical Novelty And Significance:** 2
**Recommendation:** 5
**Confidence:** 4

**Main Review:**

Overall, the paper is well-written and the proposed methodology is clear and straightforward. The models introduced in this work demonstrate state-of-the-art results. The authors compare with a wide range of previous approaches thoroughly described in Related Work.

My biggest concern with the current version of the paper is a low methodological novelty: the approach fairly similar to the previously proposed approaches and is not introducing some principally new ideas or mathematical methods. Also, the authors acknowledge that the presented distillation approach is already used for the downstream tasks.

Another concern is that, the results presented in the paper are not compared with other participants of SemEval tasks (and based on the SemEval metrics). Concerning comparison to other systems, I also suggest to compare to another simple approach for obtaining static sense embedding out of word embedding by Pelevina et al. (2017).

It would be also important to explain in more detail why the authors chose Topic Modelling as the extrinsic evaluation task. The choice of the exact three senses (k=3) is also not justified. Another concern is about the corpora the models are trained on. The authors insist that training models on the April 2010 snapshot of Wikipedia is for fair comparison with the prior multi-sense embedding models. At the same time they apply the base BERT model which is pretrained on the later version of Wikipedia.


**Summary Of The Paper:**

The paper studies the problem of word sense induction using knowledge of the large pre-trained model BERT. The authors propose a two-stage method to distill multiple word senses by using attention over the senses of a word in a context and transferring this sense information to fit multi-sense embeddings in a skip-gram-like framework.

This paper has the following contributions: a two-stage method for knowledge distillation from BERT to static word sense embeddings, a sense embeddings model and application of such embedding models to the topic modelling task. Topic modelling is used as an extrinsic evaluation.


**Summary Of The Review:**

The paper proposes fairly straightforward, yet effective approach based on BERT. Intinsic evaluation setup and the structure of the paper is more or less identical to that of Bartunov et al. (2016). The section about the extrinsic evaluation of the proposed method should be extended (adding more details about the topic modelling), but preferably another use-case of the proposed approach.

---

> ### Author Response · Authors · 2021-11-19
> **Distillation to multi-sense embeddings and evaluation methods for the model**
>
> We thank you for your well written review. Here we comment on the concerns raised in your review.
>
> Distillation is a general approach, whose performance in a given application depends on the choice of architectures. Distillation from BERT (DistilBERT, etc.) and skipgram-based multisense embeddings (MSSG, disambiguated skipgram, etc.) have been proposed before, but as far as we are aware, distilling from BERT to skipgram embeddings is novel, for either the single sense or multi-sense case.
> Thus our contribution is to propose distillation with this architecture, provide an algorithm for doing so (in particular, iterative sense disambiguation plays a significant role in the performance of this method) and empirically establishing the benefits of the proposed approach.
> Distillation is used to distill from larger BERT models to smaller BERT models. As far as we are aware, distillation from BERT has not been used to create multi-sense word embeddings.
>
> We follow Bartunov et. al (2016) in using ARI as the cluster evaluation metric rather than the F-score and V-measure used in SemEval, because the former is artificially higher for clusters with few centers while the latter is artificially higher for clusters with a large number of centers.
> AdaGram and disambiguated skip-gram, which were selected as our baselines, are state-of-the-art for word sense induction using word embeddings. In particular, Pelevina et al. 2017 uses AdaGram as a baseline, and does not outperform AdaGram.
>
> Topic modeling is a natural application as the same word can occur in different topics with different senses: e.g. star is a word in "science" in one sense and a word in "entertainment" in another sense. We expect that accounting for that difference should lead to better topic modeling.

---

### Comment · Area_Chair_qUDs · 2021-11-16
**Additional Discussion Encouraged**

Dear Reviewers,

can you please take a look at each other's reviews? Your reviews currently straddle the decision boundary and it would be good to make sure you have considered all the perspectives provided. Please update your reviews (at least to acknowledge that you have read all reviews).

Thanks,
Your Area Chair

---

### Decision · Program_Chairs · 2022-01-20

**Decision:**

Reject

**Comment:**

This paper investigates a technique for projecting contextual embeddings into static embeddings. Neither the technique is ver novel, nor are the empirical results very strong. While the reviewers did not engage in a discussion, the area chair does not see this paper reaching the quality bar of the conference.